# Cognitive Function Trajectories and Factors among Chinese Older Adults with Subjective Memory Decline: CHARLS Longitudinal Study Results (2011–2018)

**DOI:** 10.3390/ijerph192416707

**Published:** 2022-12-13

**Authors:** Chifen Ma, Mengyuan Li, Chao Wu

**Affiliations:** 1School of Nursing, Peking University, 38 Xueyuan Road, Haidian District, Beijing 100191, China; 2College of Health Services and Management, Xuzhou Kindergarten Teachers College, Xuzhou 221001, China

**Keywords:** older adults, cognitive function, trajectory, subjective memory decline, latent growth curve model

## Abstract

Older adults with subjective cognitive decline are at increased risk of future pathological cognitive decline and dementia. Subjective memory decline is an early sign of cognitive decline; preventing or slowing cognitive decline in at-risk populations remains an elusive issue. This study aimed to examine the cognitive trajectories and factors in older adults with subjective memory decline. Latent growth curve models (LGCMs) were fitted to examine the cognitive function trajectories and factors among 1465 older adults (aged 60+ years) with subjective memory decline. Data were obtained from four waves from the China Health and Retirement Longitudinal Study (CHARLS, 2011–2018), which is a large nationally representative sample of the Chinese population. The results showed that older adults with better initial cognition had a slower decline rate, which may be accelerated by advanced age, low-level education, a rapid decrease in instrumental activities of daily living (IADL) ability, and rapid increase in depression levels. This study was the first to examine the trajectories of cognitive function and its factors in a high-risk population with subjective memory decline. These findings may guide prevention approaches to tackle the issues of cognitive function decline and dementia.

## 1. Introduction

Cognitive function is an advanced function of human mental activities, which mainly reflects the ability of individual intelligence to process, store and extract information. Cognitive function involves a number of specific cognitive dimensions, including executive function/reasoning, processing speed, attention, memory, language, and visual–spatial ability/construction [1]. People often experience irreversible cognitive decline as they age. Older adults with continued cognitive decline may eventually develop dementia. Cognitive decline encompasses various cognitive dimensions including memory, numerical calculation, language, constructability, and executive function [2]. Studies show that there is significant heterogeneity in the rate of decline of certain cognitive subdomains in older adults. For example, vocabulary and general knowledge usually remain stable and even improve with the accumulation of life experience. In contrast, complex attention, processing speed, and memory may continue to decline with age [1]. Attention is individuals’ ability to focus on a specific stimulus. Late in life, people may show only a slight decline in simple attention tasks but significant declines in complex attention tasks (e.g., older adults may experience difficulty in calculating shopping bills). As one of the complex attention tasks, the numerical calculation is a complex cognitive process requiring multifactorial processes [3]. Numeracy is a learned ability related to age, education, gender, and executive function [4]. Age-related memory changes include declarative (semantic and episodic memory) and nondeclarative (implicit) memory declines. Memory decline is associated with slowed processing speed and decreased use of strategies in learning [1]. Regarding visual–spatial ability, older adults’ ability to combine separate parts into a coherent whole (visual construction skills) declines with age, whereas visuospatial ability remains stable [5].

Cognitive decline in older adults is insidious. Although many individuals show no objective evidence of cognitive impairment, they may experience subjective cognitive decline years before the abnormal performance on neuropsychological tests [6]. Subjective cognitive decline (SCD) refers to “a subjectively perceived decline in memory or non-memory domains” [7]. Currently, an increasing number of individuals are seeking medical help and advice for SCD [8]. Additionally, older individuals experiencing SCD are at an increased risk of future pathological cognitive impairment and dementia [7]. This might be since SCD is associated with episodic memory-related hippocampal atrophy, which is considered a hallmark of Alzheimer’s disease [9]. A longitudinal study on 2415 older Germans reported that the risk of progression from SCD to mild cognitive impairment (MCI) increased 20 to 60 fold [10]. Subjective memory decline (SMD), a main symptom of SCD, was related to Alzheimer’s disease (AD) biomarkers before onset and increased the risk of developing MCI and dementia [11].

In primary healthcare-related research, a growing number of public health and social science studies have focused on memory loss and its impact and social consequences. In survey-based studies, self-reported memory was crucial for the early detection of memory impairment and is usually the key to early treatment, the establishment of coping strategies, and complementary intervention measures to manage its symptoms [12]. Doctors often use self-reported memory as an informal cognitive screening method, and conduct further cognitive tests based on the results [13]. High-tech imaging and biomarkers support the notion that SCD may predict the early pathology of Alzheimer’s disease [14]. It is particularly important to describe and characterize the trajectory of age-related cognitive decline and its factors among those with a potential future risk of developing dementia. It has been reported that factors associated with cognitive function among older adults included age, sex, education levels, smoking, drinking, marital status, number of comorbidities, sleep duration, depression, and instrumental activities of daily living (IADL) [15,16,17]. Previous studies reported that subjective memory complaints were related to depression and poorer IADL ability among participants with SMD [18,19]. However, there have been limited studies on cognitive function in Chinese older adults with SMD creating a higher risk for developing dementia compared to the general population.

Most previous longitudinal studies focused on cognitive performance in general older people and the protective and risk factors of age-related cognitive decline and found that there could be subgroups with heterogeneous cognitive trajectories [20,21]. Older adults exhibit different patterns across different dimensions of cognitive function [22]. Cognitive development patterns in general populations are associated with depression, IADL, social activities, marital status, and diabetes [23]. To date, the pattern of cognitive development and the factors that predict cognitive decline in the Chinese population with SMD remains unclear.

Among the diverse trajectory models, the latent growth curve model (LGCM) is considered ideal for identifying cognitive function trajectories and their associated factors [21,22,23]. Therefore, this study used the LGCM methods and aimed to identify the cognitive trajectory and explore the protective and risk factors in Chinese people aged 60+ years with SMD. Specifically, three trajectory models investigated the trajectories of global cognitive function and each dimension, association between cognitive trajectories and baseline covariates and time-varying covariates, and association of cognitive growth trajectories with developmental trajectories of factors that showed substantial association with all cognitive dimensions across the four waves found in the second LGCM.

## 2. Materials and Methods

### 2.1. Study Design and Participants

Our study used data from four waves of the China Health and Retirement Longitudinal Study (CHARLS, 2011–2018). CHARLS is a publicly accessible database created by the National School of Development at Peking University. Further details can be found on the official website (http://charls.pku.edu.cn, accessed on 12 March 2022). The CHARLS was a large, nationally representative cohort study that focused on participants aged 45+ years, which included information regarding their social, economic, family, activity, and personal health conditions. The survey was launched in 2011 and followed up in 2013, 2015, and 2018. The sample was randomly selected from 150 counties or districts, with 450 communities or villages distributed across 28 provinces. The biomedical ethics committee approved this program (IRB00001052-11015), and all participants provided informed consent. All participants were interviewed face-to-face by trained professional researchers using structured questionnaires. Detailed sampling methods were described in Zhao et al. [24]. Figure 1 shows a flowchart of the sample selection process. 

### 2.2. Cognitive Assessment

#### 2.2.1. Subjective Memory Decline

Subjective memory decline (SMD) was identified based on one question. Participants were asked, “How would you rate your memory at present? Would you say it is excellent, very good, good, fair, or poor?”. Participants who replied fair or poor were included [25] in this study.

#### 2.2.2. Cognitive Function

Cognitive function assessment tools, derived from an adapted Chinese version of the Mini-Mental State Examination (MMSE) [26], included four dimensions: orientation, verbal episodic memory, numerical calculation, and constructability [27]. Orientation (range = 0–5) was evaluated by naming the date (day, month, and year), day of the week, and current season [28]. Verbal episodic memory (range = 0–20) was measured using immediate and delayed word recall. Participants were shown 10 Chinese nouns and asked to repeat them instantly (immediate memory) and four minutes later (delayed memory) [29]. Numerical calculation (range = 0–5) was derived by asking the participants to answer 7 subtracted from 100 five times in succession [30]. Constructability (range = 0–1) was evaluated by asking the participants to redraw a picture of two overlapped pentagons [31]. The global cognitive score (range = 0–31) was the sum of four scores [32]. A higher score indicated better global cognition and subdimensions.

### 2.3. Covariates

#### 2.3.1. Demographic and Health-Related Variables

Demographic characteristics included age (≥60 years), sex (male = 0, female = 1), education level (effect codes: illiterate = −1, literate = 0/1), smoking (no = 0, yes = 1), drinking (no = 0, yes = 1), and marital status (married with spouse present = 1, without spouse present = 0). Number of cardiovascular and cerebrovascular diseases was calculated as the sum of the dichotomous (yes = 1, no = 0) variables from the CHARLS data (which included hypertension, dyslipidemia, diabetes, heart diseases, stroke, and dementia/Parkinson’s disease). Self-reported night sleep time (hours) and napping time (minutes) were also recorded.

#### 2.3.2. Instrumental Activities of Daily Living (IADL)

The wave-1 (baseline) CHARLS only assessed the five IADL domains by asking participants the following questions: “Due to health and memory problems, are you having difficulty with the following activities? (Exclude those that you think will last for less than three months). These activities included doing housework, cooking hot meals, shopping for groceries, managing money, and taking medication.” [33]. For every question, there were four possible responses (range = 1–4), which included no difficulty, some difficulty yet still able to complete, some difficulty and need help, and unable to complete. Higher total scores (range = 5–20) indicated a lower ability to perform IADL.

#### 2.3.3. Depression

The CHARLS used the 10-item Center for Epidemiology Studies Depression Scale (CES-D-10) to assess depression in each wave. Possible responses for each item were from “Rarely or none (<1 day)” to “most or all of the time (5–7 days)” (range = 0–3). Higher total scores (range = 0–30) indicated more severe depressive symptoms. The CES-D-10 showed good internal consistency with a reliability coefficient of 0.80 in the current study. A cutoff score of 12 was used to identify depressive symptoms [34].

### 2.4. Statistical Analysis

Descriptive statistics of individual demographic and health-related variables in the four waves were summarized. Means and standard deviations (SD) for continuous variables and percentages for categorical variables were calculated.

The hypothesized latent growth curve models (LGCMs) are presented in Figure 2. Three LGCMs were fitted to identify the cognitive function trajectories and their determinants. The LGCMs used two growth factors to capture the trajectories: the intercept (I), which indicated the levels of outcomes at baseline, and slope (S), which indicated the rates of change over the follow-up period. Model A (Figure 2A): a univariate LGCM (without controlling for covariates) explored the cognitive functions’ initial level (intercept), their rate of change (slope), and the correlation between them [35]. Model B (Figure 2B): Conditional LGCM investigated the trajectories of cognitive function and its association with baseline covariates and time-varying covariates [23]. Growth factors of cognitive function (intercepts and slopes) were regressed on baseline covariates, while cognitive function at four waves of time was regressed on time-varying covariates. Model C (Figure 2C): Unconditional Parallel Process LGCM (PP-LGCM) further investigated the relationship between growth trajectories of cognitive function, IADL, and depression in the four waves. This model estimated the parameters of intercepts and slopes of each time-moving variable and the interrelations between them [36]. 

Satorra–Bentler scaled chi-squared (χ^2^) goodness-of-fit tests evaluated the model fit [37]. The model was considered acceptable when the Comparative Fit Index (CFI), Tucker–Lewis Index (TLI), Standardized Root Mean Square residual (SRMR), and Root Mean Square Error of Approximation (RMSEA) were ≥0.90, ≥0.90, ≤0.05, and <0.08, respectively [38]. Demographic data were analyzed using STATA version 17.0. A total of 1000 bootstrap samples were performed with Mplus 7.4 [39]. Missing data were processed using the full information maximum likelihood estimation method, which allowed the estimation of model parameters based on all available data and provided robust estimation when missing data were random and multivariable normal was assumed [40].

## 3. Results

### 3.1. Descriptive Statistics

Table 1 shows the socio-demographic and health statistics of the sample characteristics at baseline (2011) and the four waves (2011–2018). The baseline survey included 6369 respondents recruited in wave 1 (2011). Overall, 4903 participants were removed due to a loss of follow-up and missing information for cognitive assessment. Finally, we included 1465 participants in the cognitive trajectories analysis. The mean score for global cognition and the four dimensions showed a downward trend over time. Data on IADL and depression indicated gradual curvilinear changes, with a slight decrease in 2013 and subsequent increase in 2015 and 2018. 

### 3.2. Univariate LGCM of Cognitive Function

The results of Model A indicated that cognitive function showed a linear decline change across the four waves with significant variation across individuals. The decline rate of episodic memory and calculation ability was significantly faster compared to that of other cognitive dimensions. The higher the initial score of global cognitive function and episodic memory, the slower the gradual decline rate. 

Table 2A shows the univariate LGCM fitting results, estimating the growth trajectory of cognitive function. Model fit for the univariate LGCM of each cognitive function domain was good. The average baseline value of global cognition was 15.759 (*p* < 0.001), and the average rate of decrease for each additional year was −0.251 (*p* < 0.001). Similarly, the values of the four dimensions decreased linearly, and the trend was almost synchronized with global cognition. The decline rate of episodic memory (−0.104) and calculation (−0.095) was faster compared to that of other cognitive dimensions. Variances in the intercepts and slopes were statistically significant, which suggested significant variation across individuals in initial cognitive functions and change rates in global cognition, orientation, and verbal episodic memory. The correlation indicated a relationship between the intercept and slope. The statistically significant correlation value of global cognition and episodic memory suggested a significant association between initial score and change rates, which meant that the higher the initial score, the slower the decline.

### 3.3. Conditional LGCM

The results of Model B indicated that the initial level of cognitive function was related to age, sex, education, marital status, cardiovascular and cerebrovascular diseases, night sleeping time, depression, and IADL ability. This decline may be accelerated by advanced age, low-level education, a rapid decrease in IADL ability, and rapid increase in depression levels.

Due to missing information on the depression variable, the sample size included in model B and model C was 1426. Table 2B presents the conditional LGCM fitting results. The high fit indices showed that the model-fitting effect was good, especially for the calculation and constructability models. High fit indices indicated a high consistency between the predicted values from the hypothesized theoretical model and actual data. Associations between age and growth factors of global cognition and verbal episodic memory were significant, which indicated that the higher the age, the lower the cognitive functions at baseline on average in global cognition and episodic memory and faster the rate of decline in global cognition. Regarding gender, the results suggested that compared to males, females reported higher scores in verbal episodic memory and lower scores in calculation at baseline on average. The results indicated that the higher the level of education, higher the baseline score in each cognitive domain and slower the rate of decline in global cognition and verbal episodic memory. Moreover, night sleeping hours had a significant effect on constructability, which indicated that having a longer night sleeping duration was related to a higher initial score of constructability.

For time-varying covariates, marital status had a significant relationship with global cognition in waves 2, 3, and 4, and episodic memory in waves 2 and 4. IADL and depression were significantly associated with almost all the dimensions of cognitive function in the four waves. Accordingly, we used an unconditional parallel process LGCM to further examine the relationship between cognitive growth trajectories and developmental trajectories of IADL and depression.

### 3.4. Unconditional Parallel Process LGCM

The findings of Model C indicated that initial cognitive function was significantly associated with IADL and depression. Older adults with poorer or accelerated decline in IADL ability and rapidly increasing depression levels may experience rapid cognitive decline. Since the trends of IADL and depression showed curvilinear changes, both linear and nonlinear (quadratic curve) growth patterns were explored. Table 3 shows the associations between curve growth trajectories on parallel variables (cognitive function vs. IADL and depression) over time. The fitting results showed satisfactory goodness-of-fit (Appendix A). Appendix A showed the results of the linear models. Results showed that a higher initial IADL score was significantly associated with a lower initial cognition score and slower decline rate. There was also a significant association between the rate change of cognition and IADL, which suggested that participants with an accelerated decrease in IADL ability had a rapid cognitive decline. Regarding depression and cognitive function, results suggested that a higher initial score for depression was significantly associated with a lower initial score for global cognition and all four dimensions. Similarly, participants with accelerated increase in depression scores experienced rapid declines in global cognition, orientation, and episodic memory.

A summary of the findings regarding the trajectory hypotheses is provided in Appendix A. The attrition analyses are shown in Appendix A. Appendix A shows the univariate trajectories of cognitive function and association between cognitive trajectories and baseline covariates and time-varying covariates among participants without SMD.

## 4. Discussion

This retrospective longitudinal study examined cognitive trajectory and its protective and risk factors over eight years in a large nationally representative cohort study of Chinese older adults with SMD at potential risk of developing future dementia. We found that cognitive trajectories decreased faster with time in older adults with SMD compared to those without SMD. Older adults with SMD with higher initial global cognition and episodic memory scores showed a less rapid decline. Cognitive decline could be accelerated by advanced age, low-level education, a rapid decrease in IADL ability, and rapid increase in depression levels.

Previous studies indicated that SMD is associated with global cognitive decline over six years and may be a predictor of dementia [41]. Compared to a previous cohort study that focused on older Chinese adults in the general population from 2002 to 2014 [23], our findings showed that older individuals with SMD had a lower initial score (15.759 vs. 27.88) and rapid decline rate (−0.251 vs. −0.15) in global cognitive function. The main reason for this might be that older individuals who experienced SMD might have already experienced episodic memory-related hippocampal atrophy [9]. Imaging evidence supported that participants with SMD showed white matter integrity changes in the left hippocampus before they exhibited objective cognitive impairment [6]. Based on lower hippocampal volume, a recent study [42] suggested that SMD participants might have a higher level of homocysteine, which is a risk factor for transition from cognitively unimpaired to amnestic mild cognitive impairment. In this study, episodic memory and calculation ability declined significantly faster compared to other dimensions, which might have driven the results of global cognition. 

Episodic memory, also called autobiographical memory, is the memory of a personal experience that occurred at a specific place and time. In contrast to semantic memory, which declines later in life, episodic memory shows a lifelong decline and is associated with educational attainment [1,43]. In this study, SMD older adults’ episodic memory was significantly related to age, gender, and education. Our results indicated that women scored higher than men, which was consistent with the previous study [44]; this may be due to gender differences in sociocultural processes [44]. Mother–child conversations, for example, establish basic memory tendencies early in life and continue to influence them into adulthood [45]. However, regarding the subdomain of numerical calculation, our results showed that men scored significantly higher than women. “Calculation” is a learned ability closely related to the level of education and culture. The fact that men had more access to education than women in China over the last century, especially in rural areas, may have been an essential factor in this result.

We examined the cognitive trajectories and factors in participants with and without SMD to explore the predictors that might be unique to SMD. We found that the effect size of the factors associated with cognitive function was significantly smaller in non-SMD participants compared to in those with SMD. In those with SMD, the associations between cognitive function and age, educational level, IADL, and depression were stronger, which suggested that these factors may represent targets for intervention. Consistent with existing evidence from other longitudinal studies [32,46], our findings confirmed that factors, such as advanced age, sex, low-level education, IADL difficulty, and depression, were associated with the initial level of cognitive function. Previous longitudinal studies [22,46] on cognitive trajectories and their factors for general older adults found that smoking, alcohol consumption, higher initial levels of depression, and lower IADL ability at baseline were associated with a sharp decline in cognitive function. In this study, we did not find an association between smoking, alcohol consumption, and the development of cognitive function in participants with SMD. Nevertheless, for non-SMD individuals, the orientation dimension was significantly correlated with smoking and chronic diseases.

To examine the growth trajectory between global cognition, IADL, and depression, we conducted two PP-LGCMs. Results suggested that the rate of cognitive decline among participants with SMD was closely related to changes in IADL and depression levels. Thus, there may be a bidirectional relationship between cognitive decline and IADL dependency. Prior studies showed that cognitive decline was associated with gradual IADL disability in older adults [47]. One of the main characteristics of the differential diagnosis between MCI or SCD/SMD and dementia is the ability to maintain autonomy and independence in the activities of daily living (ADL) [48]. Thus, ADL impairment is one of the criteria used to distinguish between dementia and its prodromal symptoms. However, complex IADL function impairment may occur in older adults with MCI [49]. IADL is presented as a risk factor in the onset of MCI. IADL requires more complex neuropsychological processing than basic activities of daily living and therefore is more prone to impairment triggered by the loss of cognitive ability. Difficulties with IADL are an early sign of cognitive impairment, and using IADL as a screening tool for the diagnosis of MCI has been proved effective [50]. Our findings supported that the decline in IADL could predict cognitive decline, and its rate could predict the cognitive decline rate in older people with SMD. Similarly, emotion was a critical mediator in the association between subjective and objective memory [51], and there may also be a bidirectional relationship between cognitive decline and depression. Depression has been considered a potentially modifiable risk factor for cognitive impairment, and elevated depressive symptoms may be a prodromal syndrome of dementia [52]. A recent longitudinal study showed that middle-aged and older adults with increased depressive symptoms experienced a fastest decline in global cognitive function and episodic memory [53]. In addition, our findings that the rate of exacerbation of depressive symptoms was also associated with an accelerated decline in orientation in SMD participants extended previous literature. 

Regarding the relationship between night sleep and cognitive function, a study that focused on people over the age of 45 years showed an inverted U-shaped relationship between sleeping time and cognitive functions, which meant that inadequate (less than 4 h) or excessive (over 10 h) sleeping time per night was harmful to cognitive function [54]. In our study, a longer night sleeping time was associated with a higher score for constructability. This mechanism was also reported in related studies. For example, dreams during sleep may be closely related to memory and visuospatial ability [55], which suggested that individuals with fewer sleeping hours at night, which may lead to longer daytime sleep, might have lower visual–spatial ability [56]. Although our results showed that women had higher episodic memory and lower numerical calculation in the initial score compared to men, which was consistent with prior results [46], we did not find gender differences in the initial score and decline rate in global cognition.

These findings could help us understand the association between cognitive trajectories and factors and develop better approaches to prevent cognitive decline in older adults. Health professionals and researchers can use continued education, chronic disease management, healthy sleep promotion, and maintaining IADL and mental health as interventions to delay cognitive decline in older adults. Episodic memory and calculation ability might be the focus of cognitive promotion in older adults with SMD.

### Limitations

This study has some limitations. First, like all other longitudinal studies, attrition was a common problem due to the loss of follow-up. The main reasons for this loss could have been age and health problems. Second, this study’s cognitive function assessment tools derived from an adapted Chinese version of the MMSE, usually used to screen dementia, may not have been sensitive enough to detect subtle gradual cognitive changes in older adults [2]. Although a 10-item word list in the Chinese version of the MMSE was used to examine the episodic memory dimension, there may be a ceiling effect in other dimensions. Hence, future studies using objective memory measurements are required. Third, the assessment of some variables (e.g., sleeping time) was based on a subjective report and may be limited by the potential bias of the self-report measures. Furthermore, there was only one self-report item for SMD. The assessments of subjective memory complaints via a single self-reported question may bias the results. Self-perceptions may lack objectivity, and the validity needs improvements since a single question cannot capture the multi-dimensional structure of subjective memory, such as working, episodic, visuospatial, and semantic memory. Future studies should adapt comprehensive assessment tools such as the Subjective Memory Complaints Questionnaire (SMCQ) [57] and Multifactorial Memory Questionnaire (MMQ) [58], to enhance the validity of the assessment. Fourth, IADL has been reported to be assessed differently in men and women because of different gender roles and environmental and cultural factors [48]. IADLS, as presented in this study, is only across five domains, which may also contain a gender bias. It is recommended that future studies take gender differences into account. Finally, we fitted the univariate and conditional trajectories of cognitive function among participants without SMD and sought to explore which predictors were, to what extent, unique to SMD (Appendix A). However, the sample size of the participants without SMD was small. Hence, future research could further explore this issue using a larger sample size.

## 5. Conclusions

This study was the first to use three types of latent growth curve models (LGCMs) to examine trajectories of cognitive functions and the factors that influenced them among older Chinese adults with SMD from a large nationally representative cohort study over eight years. The findings indicated that older adults with higher initial global cognition and episodic memory scores declined less rapidly. The initial level of cognitive function was related to age, sex, education, marital status, cardiovascular and cerebrovascular diseases, night sleeping time, depression, and IADL ability. This decline may accelerate with advanced age, low-level education, a rapid decrease in IADL ability, and a rapid increase in depression level. Our outcomes have potentially significant implications for determining the characteristics of age-related cognitive decline and its determinants among populations with SMD at potential risk of developing future dementia. These cognitive risk and protective factors may guide preventive approaches to tackle the issues of increasingly globalized age-related cognitive decline and impairment.

## Figures and Tables

**Figure 1 ijerph-19-16707-f001:**
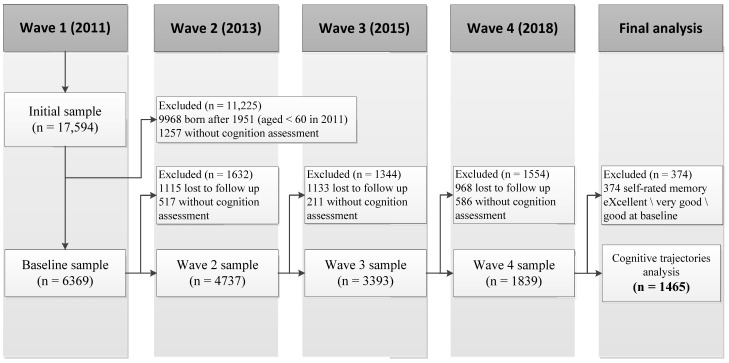
The flowchart of recruited participants for the analysis.

**Figure 2 ijerph-19-16707-f002:**
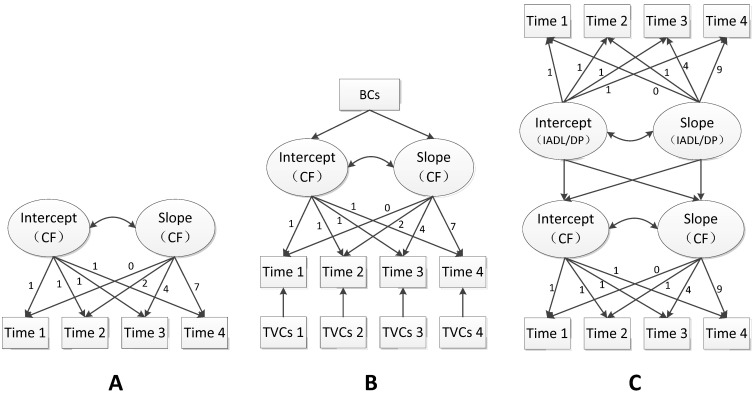
Hypothesized latent growth curve model (LGCM) of this study. CF: Cognitive Function, including Global Cognition and 5 dimensions (Orientation, Episodic Memory, Calculation, Constructability); BCs: Baseline Covariates, including age, gender, education, marriage, napping time, night sleeping time, the number of cardiovascular and cerebrovascular diseases, drinking, smoking, instrumental activities of daily living (IADL), and depression (DP); TVCs: Time-Varying Covariates, including marriage, napping time, night sleeping time, the number of chronic diseases, IADL, and depression (DP) across 4 waves. Model (**A**) Univariate LGCM, to investigate the trajectories of CF across 4 waves. Model (**B**) Conditional LGCM, to investigate the trajectories of CF and its association with BCs and TVCs. Model (**C**) Unconditional Parallel Process LGCM (PP-LGCM), to investigate the relationship between CF, IADL, and DP across 4 waves. (The data were collected at 2011, 2013, 2015, and 2018, which accounted for the time coefficients of 0, 2, 4, and 7 in model (**A**) and model (**B**). Since the trends of IADL and depression showed curvilinear changes, the quadratic curve growth patterns were explored with time coefficients of 0, 1, 4, and 9 in model (**C**)).

**Table 1 ijerph-19-16707-t001:** Descriptive statistics of sample characteristics.

**Baseline (n = 1465)**	**n/mean**	**%/SD**			**n/mean**	**%/SD**		
Age	65.47	4.56	Male		884	60.34%		
Education			Female		581	39.66%		
Illiterate	165	11.26%	Smoking (Yes)		706	48.19%		
Primary school or below	863	58.91%	Smoking (No)		759	51.81%		
Middle school	289	19.73%	Drinking (Yes)		432	37.75%		
High school or above	148	10.10%	Drinking (No)		912	62.25%		
**Four Waves**	**2011**		**2013**		**2015**		**2018**	
	**n/mean**	**%/SD**	**n/mean**	**%/SD**	**n/mean**	**%/SD**	**n/mean**	**%/SD**
Married with spouse present	1243	84.85%	1224	83.55%	1184	80.82%	1109	75.70%
Chronic diseases								
Hypertension	459	31.46%	521	35.59%	587	40.65%	747	50.99%
Dyslipidemia	200	13.91%	253	17.29%	267	18.53%	400	27.30%
Diabetes	116	7.98%	153	10.45%	166	11.40	245	16.72%
Heart disease	250	17.16%	282	19.26%	329	22.55%	430	29.35%
Stroke	34	2.32%	44	3.00%	45	3.09%	151	10.31%
Memory-related disease	27	1.85%	28	1.91%	44	3.01%	93	6.35%
Sleeping time								
Night sleeping time (hours)	6.17	1.79	6.04	1.72	6.18	1.90	6.08	2.05
Napping time (minutes)	36.25	43.17	43.29	46.59	41.82	44.76	47.35	54.86
Cognitive functions								
Global cognition	15.59	4.22	15.56	4.53	14.66	4.60	13.97	5.64
Orientation	4.09	1.09	4.09	1.17	4.04	1.16	3.84	1.16
Episodic memory	7.24	2.95	7.23	3.10	6.65	3.18	6.65	4.01
Calculation	3.50	1.74	3.50	1.72	3.26	1.80	2.87	1.89
Constructability	0.76	0.43	0.75	0.43	0.71	0.45	0.61	0.49
IADL	5.55	1.51	5.53	1.57	5.68	1.83	6.01	2.28
Depression	8.52	6.13	7.69	5.42	7.74	6.21	8.45	6.32
Depression symptoms	424	28.94%	307	20.96%	352	24.03%	418	28.59%

Instrumental activities of daily living (IADL).

**Table 2 ijerph-19-16707-t002:** Results of the univariate and conditional linear Latent Growth Curve Models (LGCM).

**A. Univariate LGCM**											
	**Mean**		**Variance**			**Fit Indexes**					
**n = 1465**	**Intercept**	**Slope**	**Intercept**	**Slope**	**Correlation**	**χ^2^**	**df**	**CFI**	**TLI**	**RMSEA**	**SRMR**
**Global cognition**	**15.759 *****	**−0.251 *****	**7.778 *****	**0.094 ****	**0.413 *****	**26.412 *****	**5**	0.988	0.985	0.054	0.025
**Orientation**	**4.134 *****	**−0.038 *****	**0.507 *****	**0.008 ****	−0.011	**25.549 *****	**5**	0.980	0.976	0.053	0.030
**Episodic memory**	**7.264 *****	**−0.104 *****	**2.714 *****	**0.038 ***	**0.161 ****	**20.759 *****	**5**	0.984	0.980	0.046	0.022
**Calculation**	**3.591 *****	**−0.095 *****	**0.839 *****	0.007	0.007	**20.226 ****	**5**	0.975	0.971	0.046	0.030
**Constructability**	**0.779 *****	**−0.022 *****	**0.048 *****	<0.001	<0.001	**15.365 ****	**5**	0.977	0.972	0.038	0.021
**B. Conditional LGCM**											
	**Global Cognition ^a^**		**Orientation ^a^**		**Episodic Memory ^a^**		**Calculation ^a^**			**Constructability ^a^**	
**n = 1426**	**β**	**SE**	**β**	**SE**	**β**	**SE**	**β**		**SE**	**β**	**SE**
**Intercept**	**5.824 *****	0.267	**5.835 *****	0.287	**4.500 *****	0.295	**3.829 *****		0.281	**3.394 *****	0.285
Intercept variance	**0.677 *****	0.037	**0.706 *****	0.041	**0.779 *****	0.043	**0.774 *****		0.120	**0.513 *****	0.099
**Slope**	**−1.142 ****	0.260	−0.265	0.207	**−0.972 ****	0.291	**−1.215 ****		0.462	−0.389	3.382
Slope variance	**0.766 *****	0.084	**0.955 *****	0.037	**0.659 *****	0.133	**0.836 *****		0.120	**0.895 ****	0.282
**Correlation**	0.255	0.183	**−0.257 *****	0.093	0.198	0.256	−0.063		0.224	0.075	0.378
**Intercept on Baseline**											
Age	**−0.157 *****	0.034	0.008	0.035	**−0.219 *****	0.041	−0.080		0.042	−0.048	0.042
Gender	0.010	0.045	−0.024	0.046	**0.123 ***	0.055	**−0.122 ***		0.056	−0.097	0.056
≤Primary school	**−0.105 *****	0.035	−0.036	0.037	**−0.133 ****	0.044	−0.037		0.045	0.008	0.045
Middle school	**0.248 *****	0.037	**0.239 *****	0.038	**0.148 ****	0.045	**0.189 *****		0.047	**0.349 *****	0.048
≥High school	**0.368 *****	0.039	**0.315 *****	0.041	**0.308 *****	0.049	**0.229 *****		0.050	**0.328 *****	0.052
Smoking status	0.028	0.041	0.007	0.043	0.008	0.051	0.065		0.052	0.040	0.052
Drinking status	−0.038	0.036	−0.027	0.038	−0.053	0.045	0.010		0.046	0.032	0.046
Marriage	−0.030	0.680	0.010	0.072	−0.058	0.089	0.047		0.089	0.026	0.095
Night sleeping time	0.054	0.053	−0.016	0.055	0.017	0.068	0.097		0.067	**0.154 ***	0.103
Napping time	−0.052	0.051	−0.032	0.053	−0.106	0.066	0.023		0.066	0.020	0.071
CCVD	−0.002	0.075	0.113	0.077	−0.131	0.097	0.051		0.098	0.154	0.103
IADL	**−0.118 ***	0.051	**−0.128 ***	0.054	−0.122	0.067	−0.018		0.067	−0.047	0.072
Depression	−0.108	0.056	**−0.129 ***	0.058	−0.038	0.074	−0.058		0.073	−0.045	0.079
**Slope on Baseline**											
Age	**−0.158 ****	0.061	−0.073	0.057	−0.130	0.070	−0.155		0.094	−0.085	0.102
Gender	0.073	0.078	−0.050	0.075	0.143	0.092	−0.114		0.118	0.015	0.130
≤Primary school	−0.023	0.062	0.011	0.059	−0.046	0.071	−0.004		0.090	−0.062	0.105
Middle school	**0.226 ****	0.068	0.061	0.061	**0.243 ****	0.080	0.159		0.102	−0.123	0.113
≥High school	**0.189 ****	0.071	−0.012	0.066	**0.271 ****	0.087	−0.011		0.111	0.101	0.119
Smoking status	−0.092	0.072	−0.118	0.069	0.055	0.082	−0.136		0.108	0.051	0.120
Drinking status	0.001	0.063	0.003	0.060	0.026	0.072	−0.048		0.093	−0.023	0.105
Marriage	0.083	0.105	−0.033	0.082	0.105	0.122	0.059		0.158	−0.054	0.179
Night sleeping time	−0.103	0.087	−0.033	0.081	−0.003	0.101	−0.261		0.139	−0.211	0.159
Napping time	0.058	0.086	0.006	0.081	0.137	0.101	−0.055		0.127	−0.017	0.145
CCVD	0.157	0.117	0.111	0.113	0.226	0.135	−0.049		0.175	−0.072	0.200
IADL	0.136	0.087	0.068	0.083	0.130	0.102	0.062		0.129	0.196	0.159
Depression	−0.013	0.096	0.053	0.091	−0.116	0.111	0.089		0.142	−0.138	0.166
**TVCs → Cognitive Functions**											
T1_(Marriage)_ → T1	0.021	0.042	0.005	0.043	0.027	0.047	−0.025		0.046	0.018	0.047
T2_(Marriage)_ → T2	**0.073 ***	0.028	0.031	0.030	**0.063 ***	0.032	0.028		0.034	0.027	0.033
T3_(Marriage)_ → T3	**0.054 ***	0.021	0.044	0.023	0.030	0.024	0.035		0.024	0.015	0.024
T4_(Marriage)_ → T4	**0.078 ****	0.025	0.011	0.028	**0.088 ****	0.027	0.042		0.030	−0.058	0.030
T1_(Night sleeping time)_ → T1	−0.033	0.037	0.009	0.038	−0.031	0.041	−0.017		0.046	−0.077	0.042
T2_(Night sleeping time)_ → T2	−0.038	0.023	−0.042	0.025	−0.026	0.026	−0.019		0.027	−0.030	0.027
T3_(Night sleeping time)_ → T3	0.006	0.020	0.020	0.022	0.001	0.023	−0.008		0.025	−0.007	0.024
T4_(Night sleeping time)_ → T4	0.005	0.020	−0.028	0.022	0.010	0.022	0.034		0.024	−0.017	0.024
T1_(Napping time)_ → T1	0.036	0.035	0.0019	0.037	0.063	0.040	−0.012		0.040	−0.021	0.041
T2_(Napping time)_ → T2	−0.022	0.022	−0.012	0.024	−0.035	0.025	0.010		0.026	0.018	0.025
T3_(Napping time)_ → T3	−0.016	0.020	−0.032	0.022	0.003	0.023	−0.023		0.025	−0.018	0.025
T4_(Napping time)_ → T4	0.011	0.020	0.017	0.024	−0.008	0.023	<0.001		0.025	0.001	0.025
T1_(CCVD)_ → T1	0.072	0.050	−0.006	0.052	**0.143 ***	0.056	−0.028		0.056	−0.059	0.056
T2_(CCVD)_ → T2	0.059	0.032	0.006	0.052	0.066	0.036	−0.046		0.038	0.111	0.037
T3_(CCVD)_ → T3	0.006	0.026	−0.030	0.028	0.041	0.029	<0.001		0.030	−0.006	0.030
T4_(CCVD)_ → T4	0.025	0.026	−0.032	0.029	0.051	0.029	0.014		0.031	−0.037	0.031
T1_(IADL)_ → T1	−0.022	0.036	−0.044	0.037	0.010	0.041	−0.017		0.041	**−0.090 ***	0.042
T2_(IADL)_ → T2	**−0.053 ***	0.002	**−0.053 ***	0.024	−0.034	0.025	−0.024		0.027	**−0.052 ***	0.026
T3_(IADL)_ → T3	−0.040	0.021	**−0.055 ***	0.023	−0.007	0.024	**−0.048 ***		0.026	**−0.091 *****	0.025
T4_(IADL)_ → T4	**−0.071 ****	0.021	**−0.071 ****	0.024	−0.030	0.024	**−0.079 ****		0.026	**−0.094 *****	0.026
T1_(Depression)_ → T1	**−0.110 ****	0.039	−0.006	0.041	**−0.160 *****	0.044	0.011		0.044	−0.064	0.045
T2_(Depression)_ → T2	**−0.134 *****	0.024	**−0.123 *****	0.026	**−0.124 *****	0.027	**−0.062 ***		0.029	**−0.078 ****	0.028
T3_(Depression)_ → T3	**−0.100 *****	0.022	**−0.110 *****	0.024	**−0.099 *****	0.025	−0.004		0.027	−0.048	0.026
T4_(Depression)_ → T4	**−0.157 *****	0.022	**−0.099 *****	0.025	**−0.158 *****	0.025	**−0.070 ****		0.027	−0.026	0.028
**Fit Indexes**											
χ^2^	98.558		**126.072 ****		**106.172 ***		75.770			75.080	
df	79		79		79		79			79	
CFI	0.992		0.970		0.981		1.000			1.000	
TLI	0.987		0.950		0.969		1.006			1.009	
RMSEA	0.013		0.020		0.016		<0.001			<0.001	
SRMR	0.010		0.010		0.010		0.008			0.007	

^a^ Standardized score. Bold: statistically significant (* *p* < 0.05, ** *p* < 0.01, *** *p* < 0.001, uncorrected). CCVD: number of Cardiovascular and Cerebrovascular Diseases; IADL: instrumental activities of daily living. A: Univariate LGCM, to investigate the trajectories of cognitive function across 4 waves. B: Conditional LGCM, to investigate the trajectories of cognitive function and its association with Baseline Covariates (BCs) including age, gender, education, marriage, napping time, night sleeping time, chronic disease, drinking, and smoking in 2011, and Time-Varying Covariates (TVCs) including marriage, napping time, night sleeping time, chronic disease, IADL, and depression across 4 waves. T1: Time 1 (2011), T2: Time 2 (2013), T3: Time 3 (2015), T4: Time 4 (2018). The symbol of “**→**” means regression relationships.

**Table 3 ijerph-19-16707-t003:** Results of the unconditional Parallel Process curvilinear Latent Growth Curve Models (PP-LGCM).

C. Unconditional PP-LGCM					
N = 1462	β	SE		β	SE
Global cognition on IADL ^a^			Orientation on IADL ^a^		
I_(IADL)_ → I_(GC)_	**−0.347 *****	0.064	I_(IADL)_ → I_(OR)_	**−0.368 *****	0.069
S_(IADL)_ → I_(GC)_	0.067	0.083	S_(IADL)_ → I_(OR)_	0.106	0.088
I_(IADL)_ → S_(GC)_	**0.364 ***	0.151	I_(IADL)_ → S_(OR)_	**0.252 ***	0.120
S_(IADL)_ → S_(GC)_	**−0.637 ****	0.184	S_(IADL)_ →S_(OR)_	**−0.417 ****	0.147
Episodic memory on IADL ^a^			Calculation on IADL ^a^		
I_(IADL)_ → I_(EM)_	**−0.260 ****	0.070	I_(IADL)_ → I_(CA)_	**−0.223 ****	0.075
S_(IADL)_ → I_(EM)_	0.001	0.088	S_(IADL)_ → I_(CA)_	0.067	0.094
I_(IADL)_ → S_(EM)_	0.244	0.155	I_(IADL)_ → S_(CA)_	0.316	0.188
S_(IADL)_ → S_(EM)_	**−0.437 ***	0.196	S_(IADL)_ → S_(CA)_	**−0.592 ***	0.249
Constructability on IADL ^a^			Global cognition on Depression ^a^		
I_(IADL)_ → I_(CO)_	**−0.427 *****	0.090	I_(DP)_ → I_(GC)_	**−0.408 *****	0.043
S_(IADL)_ → I_(CO)_	**0.243 ***	0.113	S_(DP)_ → I_(GC)_	0.178	0.117
I_(IADL)_ → S_(CO)_	0.625	0.326	I_(DP)_ → S_(GC)_	−0.054	0.115
S_(IADL)_ → S_(CO)_	**−0.891 ***	0.428	S_(DP)_ → S_(GC)_	**−0.868 ****	0.286
Orientation on Depression ^a^			Episodic memory on Depression ^a^		
I_(DE)_ → I_(OR)_	**−0.349 *****	0.047	I_(DE)_ → I_(EM)_	**−0.383 *****	0.050
S_(DE)_ → I_(OR)_	0.172	0.126	S_(DE)_ → I_(EM)_	0.188	0.133
I_(DE)_ → S_(OR)_	0.037	0.093	I_(DE)_ → S_(EM)_	−0.102	0.129
S_(DE)_ → S_(OR)_	**−0.559 ***	0.239	S_(DE)_ → S_(EM)_	**−0.880 ***	0.339
Calculation on Depression ^a^			Constructability on Depression ^a^		
I_(DE)_ → I_(CA)_	**−0.223 *****	0.049	I_(DE)_ → I_(CO)_	**−0.358 *****	0.051
S_(DE)_ → I_(CA)_	0.074	0.123	S_(DE)_ → I_(CO)_	0.088	0.130
I_(DE)_ → S_(CA)_	0.005	0.106	I_(DE)_ → S_(CO)_	0.122	0.136
S_(DE)_ → S_(CA)_	−0.455	0.278	S_(DE)_ → S_(CO)_	−0.412	0.348

^a^ Standardized score. Bold: statistically significant (* *p* < 0.05, ** *p* < 0.01, *** *p* < 0.001). I: Intercept; S: Slop; GC: Global Cognition; IADL: Instrumental Activities of Daily Living; DP: Depression; OR: Orientation; EM: Episodic Memory; CA: Calculation; CO: Constructability. C: Unconditional Parallel Process LGCM (PP-LGCM), to investigate the relationship between global cognition, IADL, and depression across 4 waves.

## Data Availability

CHARLS is a publicly accessible database created by the National School of Development at Peking University (http://charls.pku.edu.cn, accessed on 12 March 2022).

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
