# Peer review of "Cognitive Function Trajectories and Factors among Chinese Older Adults with Subjective Memory Decline: CHARLS Longitudinal Study Results (2011–2018)"

_ijerph, 2022, doi:10.3390/ijerph192416707_

Round 1
Reviewer 1 Report
This manuscript presents sound results based on an extensive analysis. I only have one minor comment about the way the subjective memory complaint was assessed. I would say that evaluating it via a single question appears fairly problematic, and I wuld suggest to the authors to elaborate more on that in the limitation section, even though they already included it.
Furthermore, why did the authors include in the study those participants that indicated a "poor" or "fair" memory ability? Why not including only the "poor" group?
Author Response
We sincerely thank you for the valuable comments and suggestions that improve our manuscript. Below, we numbered and addressed the comments point-by-point. Our responses to each comment are in standard font, and the revised text in the manuscript is in italic font.
- This manuscript presents sound results based on an extensive analysis. I only have one minor comment about the way the subjective memory complaint was assessed. I would say that evaluating it via a single question appears fairly problematic, and I would suggest to the authors to elaborate more on that in the limitation section, even though they already included it.
Response: Thanks for your professional review work. According to your suggestion, we have elaborated on this issue in the limitation section.
“Furthermore, there was only one self-report item for SMD. The assessments of subjective memory complaints via a single self-reported question may bias the results. Self-perceptions may lack objectivity, and the validity needs improvements since a single question cannot capture the multi-dimensional structure of subjective memory, such as working, episodic, visuospatial, and semantic memory. Future studies should adopt comprehensive assessment tools, such as the Subjective Memory Complaints Questionnaire (SMCQ) [57] and Multifactorial Memory Questionnaire (MMQ) [58], to enhance the validity of the assessment.” (Page 14, lines 400-405)
- Furthermore, why did the authors include in the study those participants that indicated a "poor" or "fair" memory ability? Why not including only the "poor" group?
Response: Thanks for your comments. The definition of the participants with self-report memory loss was based on previously published literature using the CHARLS database. In their reports, “Those who reported ‘fair’ or ‘poor’ were recorded as having cognitive complaints.” We have added this reference to the revised manuscript.
“Subjective memory decline (SMD) was identified based on one question. Participants were asked, “How would you rate your memory at present? Would you say it is excellent, very good, good, fair, or poor?” Participants who replied fair or poor were included [25] in this study.” (Page 3, lines 118-121)
- Xu W, Bai A, Huang X, Gao Y, Liu L. Association Between Sleep and Motoric Cognitive Risk Syndrome Among Community-Dwelling Older Adults: Results From the China Health and Retirement Longitudinal Study. Front Aging Neurosci. 2021 Nov 5;13:774167. doi: 10.3389/fnagi.2021.774167. PMID: 34867301; PMCID: PMC8641045.
Reviewer 2 Report
Thank you very much for allowing me to read this interesting manuscript. I appreciate the interest of researchers in trying to investigate Cognitive Function Trajectories and Factors among Chinese Older Adults with Subjective Memory Decline: CHARLS Longitudinal Study Results (2011-2018). Nevertheless, there are two major theoretical issues with the paper that make the results less credible: cognitive dimensions and IADL report. Improvements are required to the current iteration before publication can be recommended. Please see the below comments and recommendations
“Cognitive domains” are presented in a confusing way throughout the manuscript, particularly in the introduction. Reference is made to "numerical calculation" but should be further explained. “Calculation” is a learned ability, not all adults have developed it in the same way during their lives, so the level of education and culture is strongly related. Although cognition is often presented as a cognitive dimension and is part of many neurocognitive tests, it is usually related to other abilities such as working memory and executive functions. In accordance with the DMS-5 there are 6 cognitive dimensions (and subdomains) learning, short and long-term memory, social cognition, language, perceptual motor, complex attention, or executive function. I recommend the authors to improve the background of the cognitive dimensions and incorporate it into their discussion.
Regarding IADLS several details, instrumental activities of daily living (IADL) refer to tasks that are necessary for living in community independently, many activities are framed as part of IADL.
Firstly, one of the main characteristics of people with MCI or SCD/SMD in the differential diagnosis with dementia is that they must maintain autonomy and independence in IADL (is not mentioned during the manuscript). IADL is presented as a risk factor in the onset of MCI, the relationship is more complex, actually a loss in cognitive ability is what would motivate the loss of autonomy, although there are subtle nuances. This should be part of your discussion.
IADLS as presented in CHARLS is only across 5 domains, which may also contain a gender bias. They should incorporate this into the discussion and limitations of the study.
Minor.
Table 1. a category under "drinking" is missing
Table 2. why is n=1426 in the LGCM?
Line 53-57 is not clear..
Line 186-187. Explain better, the baseline in 2011 according to figure1 is 6369.
Material presented in Chinese is not easily accessible due to language barriers.
Author Response
We appreciate your professional review of our article. Your suggestions have helped us improve our article. Below, we numbered and addressed the comments point-by-point. Our responses to each comment are in standard font, and the revised text in the manuscript is in italic font.
- “Cognitive domains” are presented in a confusing way throughout the manuscript, particularly in the introduction. Reference is made to "numerical calculation" but should be further explained. “Calculation” is a learned ability, not all adults have developed it in the same way during their lives, so the level of education and culture is strongly related. Although cognition is often presented as a cognitive dimension and is part of many neurocognitive tests, it is usually related to other abilities such as working memory and executive functions. In accordance with the DMS-5 there are 6 cognitive dimensions (and subdomains) learning, short and long-term memory, social cognition, language, perceptual motor, complex attention, or executive function. I recommend the authors to improve the background of the cognitive dimensions and incorporate it into their discussion.
Response: Thanks for your suggestions. We have improved the background section to explain the cognitive dimensions and incorporated them in the discussion section.
In the background section:
“Cognitive function is an advanced function of human mental activities, which mainly reflects the ability of individual intelligence to process, store and extract information. Cognitive function involves a number of specific cognitive dimensions, including executive function/reasoning, processing speed, attention, memory, language, and visual-spatial ability [1]. People often experience irreversible cognitive decline as they age. Older adults with continued cognitive decline may eventually develop dementia. Cognitive decline encompasses various cognitive dimensions including memory, numerical calculation, language, constructability, and executive function [2]. Studies show that there is significant heterogeneity in the rate of decline of certain cognitive subdomains in older adults. For example, vocabulary and general knowledge usually remain stable and even improve with the accumulation of life experience. In contrast, complex attention, processing speed, and memory may continue to decline with age [1]. Attention is individuals’ ability to focus on a specific stimulus. Late in life, people may show only a slight decline in simple attention tasks but significant declines in complex attention tasks (e.g., older adults may experience difficulty in calculating shopping bills). As one of the complex attention tasks, the numerical calculation is a complex cognitive pro-cess requiring multifactorial processes [3]. Numeracy is a learned ability related to age, education, gender, and executive function [4]. Age-related memory changes include declarative (semantic and episodic memory) and nondeclarative (implicit) memory declines. Memory decline is associated with slowed processing speed and decreased use of strategies in learning [1]. Regarding visual-spatial ability, older adults’ ability to combine separate parts into a coherent whole (visual construction skills) decline with age, whereas visuospatial ability remains stable [5].” (Page 1, lines 28-50)
“Cognitive decline in older adults is insidious. Although many individuals show no objective evidence of cognitive impairment, they may experience subjective cognitive decline years before the abnormal performance on neuropsychological tests.” (Page 1, line 51)
The following references were added:
- Harada, C.N., M.C. Natelson Love, and K.L. Triebel, Normal cognitive aging. Clin Geriatr Med, 2013. 29(4): p. 737-52.
- Gallistel, C.R., Finding numbers in the brain. Philos Trans R Soc Lond B Biol Sci, 2017. 373(1740).
- Delazer, M., G. Kemmler, and T. Benke, Health numeracy and cognitive decline in advanced age. Neuropsychol Dev Cogn B Aging Neuropsychol Cogn, 2013. 20(6): p. 639-59.
5.Howieson, D.B., et al., Neurologic function in the optimally healthy oldest old. Neuropsychological evaluation. Neurology, 1993. 43(10): p. 1882-6.
In the discussion section:
“In this study, episodic memory and calculation ability declined significantly faster compared to other dimensions, which might have driven the results of global cognition.” (Page 12, lines 316)
“Episodic memory, also called autobiographical memory, is the memory of a personal experience that occurred at a specific place and time. In contrast to semantic memory, which declines later in life, episodic memory shows a lifelong decline and is associated with educational attainment [1, 43]. In this study, SMD older adults’ episodic memory was significantly related to age, gender, and education. Our results indicated that women scored higher than men, which was consistent with the previous study [44]; this may be due to gender differences in sociocultural processes [44]. Mother-child conversations, for example, establish basic memory tendencies early in life and continue to influence them into adulthood [45]. However, regarding the subdomain of numerical calculation, our results showed that men scored significantly higher than women. “Calculation” is a learned ability closely related to the level of education and culture. The fact that men had more access to education than women in China over the last century, especially in rural areas, may have been an essential factor in this result.” (Page 12, lines 319-331)
The following references were added:
43.Rönnlund, M., et al., Stability, growth, and decline in adult life span development of declarative memory: cross-sectional and longitudinal data from a population-based study. Psychol Aging, 2005. 20(1): p. 3-18.
44.Grysman, A., Gender and gender typicality in autobiographical memory: A replication and extension. Memory, 2018. 26(2): p. 238-250.
45.Jack, F., et al., Maternal reminiscing style during early childhood predicts the age of adolescents' earliest memories. Child Dev, 2009. 80(2): p. 496-505.
- Regarding IADLS several details, instrumental activities of daily living (IADL) refer to tasks that are necessary for living in community independently, many activities are framed as part of IADL.
Firstly, one of the main characteristics of people with MCI or SCD/SMD in the differential diagnosis with dementia is that they must maintain autonomy and independence in IADL (is not mentioned during the manuscript). IADL is presented as a risk factor in the onset of MCI, the relationship is more complex, actually a loss in cognitive ability is what would motivate the loss of autonomy, although there are subtle nuances. This should be part of your discussion.
IADLS as presented in CHARLS is only across 5 domains, which may also contain a gender bias. They should incorporate this into the discussion and limitations of the study.
Response: Thanks for your professional advice. According to your suggestions, we have added the following text to the Discussion and limitations section:
In the Discussion section:
“One of the main characteristics of the differential diagnosis between MCI or SCD/SMD and dementia is the ability to maintain autonomy and independence in the activities of daily living (ADL) [48]. Thus, ADL impairment is one of the criteria used to distinguish between dementia and its prodromal symptoms. However, complex IADL function impairment may occur in older adults with MCI [49]. IADL is presented as a risk factor in the onset of MCI. IADL requires more complex neuropsychological processing than basic activities of daily living and therefore is more prone to impairment triggered by the loss of cognitive ability. Difficulties with IADL are an early sign of cognitive impairment; and using IADL as a screening tool for the diagnosis of MCI has been proved effective.” (Page 13, lines 352-361)
In the limitation section:
“Fourth, IADL has been reported to be assessed differently in men and women because of different gender roles and environmental and cultural factors [48]. IADLS, as presented in this study, is only across five domains, which may also contain a gender bias. It is recommended that future studies take gender differences into account.” (Page 14, lines 407-411)
The following references are added:
- Hesseberg, K., et al., Disability in instrumental activities of daily living in elderly patients with mild cognitive impairment and Alzheimer's disease. Dement Geriatr Cogn Disord, 2013. 36(3-4): p. 146-53.
- Jekel, K., et al., Mild cognitive impairment and deficits in instrumental activities of daily living: a systematic review. Alzheimers Res Ther, 2015. 7(1): p. 17.
Minor:
- Table 1. a category under "drinking" is missing
Response: Thanks for your reminding, we have readjusted Table 1. (Page 6, line 214)
- Table 2. why is n=1426 in the LGCM?
Response: In model B and Model C, as 39 pieces of depression variable were missing, there were n=1426 involved in model fitting. We have explained it in the revised manuscript.
“Due to missing information on the depression variable, the sample size included in model B and model C was 1426.” (Page 7, lines 241-242)
- Line 53-57 is not clear.
Response: Thanks, we have modified this sentence to make it clear.
“It has been reported that factors associated with cognitive function among older adults included age, sex, education levels, smoking, drinking, marital status, number of comorbidities, sleep duration, depression, and instrumental activities of daily living (IADL)” (Page 2, lines 74-77)
- Line 186-187. Explain better, the baseline in 2011 according to figure1 is 6369.
Response: Thank you for your suggestion. We have explained it in the revised manuscript as follows:
“The baseline survey included 6369 respondents recruited in wave 1 (2011). 4903 participants were removed due to a loss of follow-up and missing information for cognitive assessment. Finally, we included 1465 participants in the cognitive trajectories analysis.” (Page 5, lines 207-210)
- Material presented in Chinese is not easily accessible due to language barriers.
Response: Any material in CHARLS database is available in English. Please refer to the link at http://charls.pku.edu.cn/en/.
Round 2
Reviewer 2 Report
The Authors provided satisfactory revisions to the article. Thank you for responding so comprehensively to my comments ​and improving theoretical issues